# TMX5/TXNDC15, a natural trapping mutant of the PDI family is a client of the proteostatic factor ERp44

Tatiana Soldà[1,*], Carmela Galli[1,*], Concetta Guerra[1] , Carolin Hoefner[1,2], Maurizio Molinari[1,3]

The ER is the organelle of nucleated cells that produces lipids, sugars, and proteins. More than 20 ER-resident members of the protein disulfide isomerase (PDI) family regulate formation, isomerization, and disassembly of covalent bonds in newly synthesized polypeptides. The PDI family includes few membrane-bound members. Among these, TMX1, TMX2, TMX3, TMX4, and TMX5 belong to the thioredoxin-related trans-membrane (TMX) protein family. TMX5 is the least-known member of the family. Here, we establish that TMX5 covalently engages via its active site cysteine residue at position 220 a subset of secretory proteins, mainly single- and multi-pass Golgi-resident polypeptides. TMX5 also interacts non-covalently, and covalently, via non-catalytic cysteine residues, with the PDI family members PDI, ERp57, and ERp44. The association between TMX5 and ERp44 requires formation of a mixed disulfide between the catalytic cysteine residue 29 of ERp44 and the non-catalytic cysteine residues 114 and/or 124 of TMX5 and controls the ER localization of TMX5 in pre-Golgi compartments. Thus, TMX5 belongs to the family of proteins including Ero1α, Ero1β, Prx4, ERAP1, and SUMF1 that operate in pre-Golgi compartments but lack localization sequences required to position themselves and rely on ERp44 engagement for proper inter-compartmental distribution.

## Introduction

The protein disulfide isomerase superfamily includes more than 20 ER-resident members that ensure oxidative folding of newly synthesized polypeptides and participate in clearance of misfolded proteins from the ER lumen (1, 2, 3). A few of them, TMX1, TMX2, TMX3, TMX4, and TMX5, are anchored at the ER membrane (4). TMX1 (TXNDC1) is a reductase associated with mitochondria–ER contact sites. It regulates calcium flux and is involved in biogenesis and clearance of membrane-associated polypeptides (5, 6, 7, 8, 9, 10, 11). TMX2 (TXNDC14) has the active site unconventionally located at the cytoplasmic site of the ER membrane. TMX2 regulates sul-fenylation of mitochondrial proteins at mitochondria–ER contact sites to control calcium flux. It also localizes to the nuclear envelope, where it supports the nuclear import of select cargo proteins (4, 5, 12 Preprint, 13, 14). TMX3 (TXNDC10) has a canonical CGHC active site sequence, which corresponds to the catalytic site of PDI. It has oxidase activity in vitro, and its b' domain engages the oxidoreductase's clients. However, its biological role has not been established (4, 15, 16). TMX4 (TXNDC13) is a reductase located in the ER and in the nuclear envelope, where it regulates the disassembly of disulfide-bonded LINC complexes to promote selective clearance of excess outer nuclear membrane portions during recovery from ER stress (4, 17, 18, 19, 20).

TMX5 (TXNDC15) is the least-known member of the TMX family. It is a single-span type I protein of 360 amino acids with a large, hexa-glycosylated luminal domain and a short C-terminal cyto-plasmic tail lacking canonical ER retention signals. TMX5 has 5 cysteine residues (C in a single-letter code, green, Fig 1). The cysteine at position 220 (red, Fig 1) is embedded in a non-canonical CRFS active site that defines TMX5 as a natural trap-ping mutant protein (21). So far, no biological role has been ascribed to TMX5, but mutations in the TMX5 gene have been associated with the development of the Meckel–Gruber syndrome, a rare perinatally lethal autosomal recessive disease caused by defective ciliogenesis (22, 23, 24).

Here, we identify the endogenous client proteins trapped in mixed disulfides with the active site cysteine residue 220 of TMX5. We report that the endogenous PDI family members ERp57, PDI, and ERp44 associate covalently with non-catalytic TMX5 cysteine resi-dues. We characterize the association between TMX5 and the proteostatic factor ERp44 that controls the ER-to-Golgi cycling of proteins lacking ER retention or retrieval motifs (25). The associ-ation involves the TMX5's cysteine residues 114 and 124 and the ERp44 active site cysteine residue 29 and contributes to the ap-propriate subcellular localization of TMX5.

[1]Università della Svizzera italiana, Institute for Research in Biomedicine, Bellinzona, Switzerland    [2]Department of Biology, Swiss Federal Institute of Technology, Zurich, Switzerland  [3]School of Life Sciences, École Polytechnique Fédérale de Lausanne, Lausanne, Switzerland

Correspondence: maurizio.molinari@irb.usi.ch
*Tatiana Soldà and Carmela Galli contributed equally to this work

| 1 | MVPAAGRRPPRVMRLLGWWQVLLWVLGLPVRGVEVAEESGRLWSEEQPAH | 50 |
| 51 | PLQVGAVYLGEEELLHDPMGQDRAAEEANAVLGLDTQGDHMVMLSVIPGE | 100 |
| 101 | AEDKVSSEPSGVT**C**GAGGAEDSR**C**NVRESLFSLDGAGAHFPDREEEYYTE | 150 |
| 151 | PEVAESDAAPTEDS*N*NTESLKSPKVN**C**EER*N*ITGLE*N*FTLKIL*N*MSQDLM | 200 |
| 201 | DFLNP*N*GSD**C**TLVLFYTPW**C**RFSASLAPHFNSLPRAFPALHFLALDASQH | 250 |
| 251 | SSLSTRFGTVAVPNILLFQGAKPMARFNHTDRTLETLKIFIF*N*QTGIEAK | 300 |
| 301 | KNVVVTQADQIGPLPSTLIKSVD<u>WLLVFSLFFLISFIMYATI</u>RTESIRWL | 350 |
| 351 | IPGQEQEHVE | |

**Figure 1. TMX5/TXNDC15 features. TMX5 is a type I glycoprotein (the transmembrane domain is underlined).** The CRFS unconventional active site and the active site cysteine residue at position 220 are shown in red. Asparagine residues (N in a single-letter code, in N-X-S/T consensus sequences for N-glycosylation) are shown in blue. The other 4 cysteine residues C114, C124, C177, and C210 whose role is established in this study are shown in green.

# Results

### TMX5 localizes in the ER and the Golgi compartments

The presence of 6 N-linked glycans offers the possibility to establish whether TMX5 is retained in the ER or traffics to the Golgi compartment by observing changes in the electrophoretic mobility of the polypeptide and by submitting it to a standard Endoglucanase H (EndoH) sensitivity assay (26, 27). In fact, upon arrival in the medial Golgi compartment, resident glycosyltransferases modify the glycoprotein's N-glycans resulting in an increase in the polypeptide's apparent molecular weight. Moreover, the unmodified N-glycans of proteins retained in the ER are cleaved off by EndoH, whereas the complex glycans generated in the medial Golgi compartment become resistant to EndoH cleavage and can only be removed by PNGaseF (26, 27). The TMX5 immunoisolated from lysates of mammalian cultured cells (human embryonic kidney [HEK]293 cells in Fig 2A and B) exhibits heterogeneous electrophoretic mobility (green line, Fig 2A, lane 2). This is a symptom of complex glycosylation of a fraction of TMX5, which occurs when a glycoprotein escapes ER retention and is transported to the Golgi compartment. To confirm this, we exposed TMX5 to EndoH treatment that removes oligosaccharides only from the fraction of glycoproteins that is retained in the ER. The EndoH treatment does not change the electrophoretic mobility of the fraction of TMX5 above the 66 kD of apparent molecular weight (Fig 2A, lanes 3 and 4, arrow Res). In contrast, the major fraction of TMX5 (lane 3, arrow Sens) is de-glycosylated upon EndoH treatment and runs at the level of the TMX5 protein exposed to PNGaseF (Fig 2A, lane 5) that removes simple and complex N-glycans from polypeptide chains (26). Two independent experiments confirm that between 16% and 19% of TMX5 displays EndoH-resistant oligosaccharides; thus, it reaches the medial Golgi complex (Fig 2B, lanes 1, 2). Notably, complex glycosylation is not observed for TMX3 and TMX4 that are N-glycosylated members of the PDI family as well but, in contrast to TMX5, possess C-terminal KKXX retention sequences that efficiently retain them in the ER or in the contiguous nuclear envelope membrane (4, 10, 11, 17).

In addition to the active site Cys$_{220}$, TMX5 has 4 other cysteine residues at positions 114, 124, 177, and 210 (Figs 1 and 2C). Individual mutations of Cys$_{220}$, Cys$_{114}$, or Cys$_{124}$ do not affect the cycling of TMX5 between ER and Golgi as shown by the unchanged fraction of complex N-glycans displayed by the proteins, with values ranging from 15% to 21% (Fig 2B, lanes 3–8). In contrast, mutation of Cys$_{177}$ (lanes 9 and 10) or of Cys$_{210}$ (lanes 11 and 12) results in retention of the proteins in the ER as shown by the lack of the complex glycosylated immunoreactive polypeptide (Fig 2B, lanes 10 and 12). We attribute the ER retention of TMX5$_{C177A}$ and TMX5$_{C210A}$ to their misfolding. In fact, the proximity of Cys$_{177}$ and of Cys$_{210}$ in the TMX5's architecture (Fig 2C) hints at the covalent association of these two cysteine residues in a structural intramolecular disulfide bond that stabilizes the native TMX5.

Consistent with the results of the EndoH assay (Fig 2B), the analyses by confocal laser scanning microscopy (CLSM) of the subcellular localization of TMX5 and of the 5 cysteine mutants in MEFs (Fig 2D–I) show that TMX5 (Fig 2D), TMX5$_{C220A}$ (Fig 2E), TMX5$_{C114A}$ (Fig 2F), and TMX5$_{C124A}$ (Fig 2G) extensively co-localize with the ER marker CNX and with the Golgi marker Giantin (Fig 2J–M). TMX5$_{C177A}$ (Fig 2H) and TMX5$_{C210A}$ (Fig 2I) are retained in the ER and do not traffic to the Golgi compartment (Fig 2N and O).

### TMX5 is a natural trapping protein that engages endogenous polypeptides in mixed disulfides

We previously assessed the client specificity of the TMX family members in cellular localization (10, 17). To this end, we generated the client-trapping mutant forms of TMX1 (10), TMX3, and TMX4 (17) by mutating the C-terminal cysteine of their CXXC active site motif to alanine to substantially delay release of the endogenous clients from the oxidoreductases (28). This operation is unnecessary for TMX5 that has an unconventional CRFS active site that defines TMX5 as a natural trapping protein (21). The endogenous proteins that associate covalently with TMX5 via a disulfide bond were identified upon co-immunoprecipitation and mass spectrometry (MS) (10, 17). Briefly, a V5-tagged version of TMX5 was expressed in HEK293 cells. After immunoisolation of TMX5 from cell lysates with anti-V5 antibodies, the immunocomplexes were separated in non-reducing (Fig 3A, NRed) and reducing SDS–polyacrylamide gels (Fig 3A, Red), which were subsequently silver-stained. The disulfide-bonded complexes between TMX5 and endogenous polypeptides have higher molecular weight than TMX5 itself (arrow in Fig 3A) and

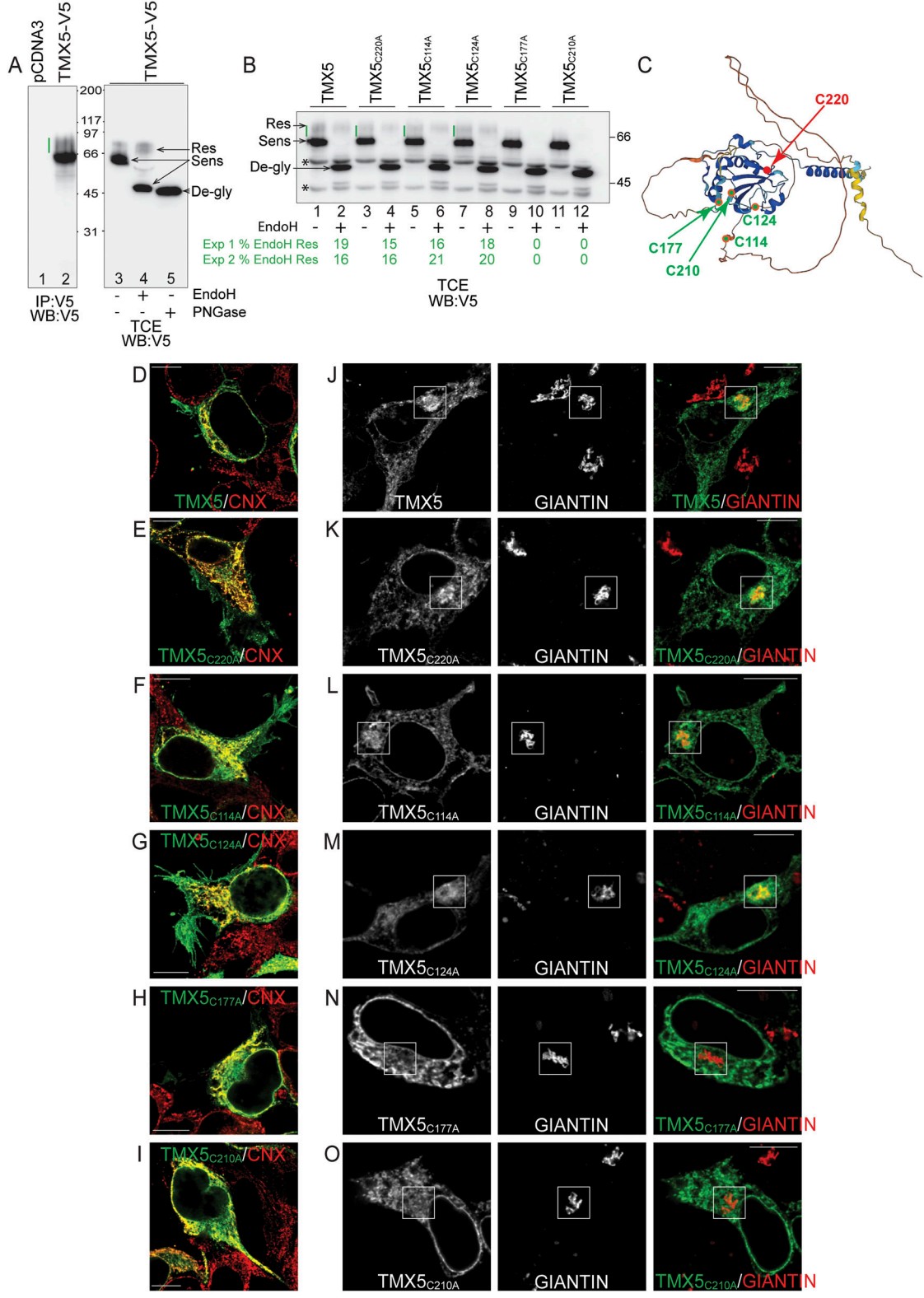

**Figure 2. TMX5 localizes in the ER and Golgi.**
**(A)** TMX5 separated in a reducing gel (lane 2). The green line shows the complex glycosylated fraction of TMX5. This fraction is resistant to EndoH cleavage (lane 4, Res). Most of the TMX5 is sensitive to EndoH treatment as shown by the shift in gel mobility (lanes 3 and 4, Sens), which corresponds to the TMX5 de-glycosylated upon exposure to PNGaseF (lane 5, De-gly). **(B)** Assessment of EndoH-resistant N-glycans of TMX5 and TMX5 cysteine mutants. The EndoH-resistant, EndoH-sensitive, and EndoH–de-glycosylated TMX5 are shown with arrows. Asterisks show cross-reacting polypeptide bands. The value of two biological replicates is shown in green. **(C)** Predicted

therefore localize in the region of the non-reducing gel shown with a red rectangle (Fig 3A, NRed). Upon reduction, disulfide-bonded complexes are disassembled, and the endogenous polypeptides that participate in the intermolecular disulfide-bonded complexes with TMX5 will disappear from the corresponding region of the reducing gel (shown with the blue rectangle in Fig 3A, Red), if their molecular weight is lower than 66 kD (i.e., the apparent electrophoretic mobility of TMX5). The regions of the gel enclosed in the red and blue rectangles were cut and processed for MS as summarized in the Materials and Methods section. The table in Fig 3B lists the endogenous polypeptides identified in the non-reducing red rectangle (column TMX5-NRed), and absent in the reducing blue rectangle (column TMX5-R). As expected, the bait TMX5 is present in both gel portions.

Among the cellular proteins that form mixed disulfides with TMX5, the major interactors are three members of the PDI superfamily (PDI [PDIA1], ERp44, and ERp57 [PDIA3]) that presumably play a role in the regulation of the function and of the intracellular localization of TMX5 (see below). Moreover, TMX5 engages in mixed disulfides with a few secretory proteins and single- and multipass ER, Golgi, and plasma membrane proteins (Fig 3B) that we define as "clients" and will analyze in a follow-up study.

### Mutation of the TMX5 active site cysteine 220 substantially reduces engagement of endogenous proteins in mixed disulfides, leaving unaffected the association of TMX5 with endogenous PDI, ERp57, and ERp44

V5-tagged TMX5 forms mixed disulfides with several endogenous proteins (Fig 3A, NRed, Fig 3B). The engagement of TMX5 in intermolecular disulfide-bonded complexes is confirmed by the separation under non-reducing conditions of TMX5-containing complexes and their detection in Western blot with the anti-V5 antibody (Fig 3C, lane 2, shown with $a$). The disulfide-bonded complexes are disassembled exposing the cell lysate to the reducing agent DTT (Fig 3A, Red, Fig 3C, lane 5). The formation of mixed disulfides with endogenous proteins is substantially reduced, as expected, upon replacement of the catalytic cysteine 220 of TMX5 with an alanine residue (Fig 3C, lane 3). Notably, $TMX5_{C220A}$ still engages endogenous proteins in mixed disulfides (Fig 3C, lane 3, shown with $b$).

The immunoisolation of V5-$TMX5_{C220A}$ from HEK293 cell lysates captures disulfide-bonded endogenous PDI (Fig 3D, lane 3, $c$, to be compared with the PDI co-immunoisolated with the wild-type TMX5 in lane 2), disulfide-bonded endogenous ERp57 (Fig 3E, lane 3, $d$), and disulfide-bonded endogenous ERp44 (Fig 3F, lane 3, $e$). These covalent complexes disappear upon reduction (Fig 3D–F, lanes 6), and the signals collapse into the monomeric forms of PDI (Fig 3D, $f$),

ERp57 (Fig 3E, $g$), or ERp44 (Fig 3F, $h$). Notably, PDI and ERp57 (not ERp44) also associate non-covalently with TMX5 as shown by their monomeric forms that co-precipitate both with TMX5 and with the $TMX5_{C220A}$ (shown with $i$ and $j$, respectively, in Fig 3D and E, lanes 2–3).

### The covalent association of ERp44 with TMX5 relies on ERp44's cysteine residue 29

The finding that endogenous ERp44 is a major interacting partner of TMX5 (Fig 3B and F) is relevant. ERp44 is an unconventional member of the PDI superfamily. ERp44 cycles between ER and Golgi to ensure the retention of key factors designed to act in pre-Golgi compartments but lacking the ER retrieval sequence (29, 30) including Ero1$\alpha$, Ero1$\beta$, Prx4, ERAP1, and SUMF1 (31). The ER retention function of ERp44 is exerted by interacting with clients via the catalytic site cysteine residue at position 29 (30). Notably, TMX5 shares the absence of an ER localization motif with all the ERp44 clients identified so far. To verify whether the active site $Cys_{29}$ of ERp44 is involved in the association with TMX5, HA-tagged ERp44 or $ERp44_{C29S}$ was co-expressed in HEK293 cells with V5-tagged TMX5. Complexes were immunoisolated from detergent lysates with anti-V5 antibodies and separated under non-reducing or reducing conditions to verify the presence of HA-tagged ERp44 and $ERp44_{C29S}$ by WB (Fig 4A and B, respectively). Analyses of the WB confirm that ERp44 engages TMX5 in mixed disulfides (arrow, lane 2, Fig 4B), whereas $ERp44_{C29S}$ does so with much reduced magnitude (Fig 4B, lane 3).

Disassembly of the complexes under reducing conditions reveals the ERp44 component of the mixed disulfides (arrow, lane 5, Fig 4C). The composition of the complexes was confirmed by their immunoisolation with an anti-HA antibody to capture the ERp44 component, and detection of the TMX5 component in WB with an anti-V5 antibody (Fig 4D and E, lanes 2 and 5). Notably, the TMX5 that co-precipitates with ERp44 (Fig 4E, lane 5) lacks the slow-migrating, complex glycosylated polypeptide fraction that cycles in the Golgi compartment shown with green lines in Figs 2A and B and 3A and C. Thus, ERp44 associates covalently via its active site $Cys_{29}$ with TMX5 located in the ER or in a pre-medial Golgi compartment, where TMX5's N-glycans would be processed to complex forms (i.e., before the $\alpha$3- and $\alpha$6-linked mannose residues of TMX5's N-glycans are substituted with GlcNAc moieties to render them resistant to cleavage by EndoH) (26, 27).

### The association with ERp44 retains TMX5 in the ER and reduces engagement of endogenous clients

The competence of ERp44 to form mixed disulfides with TMX5 is confirmed by the presence of immunoreactive TMX5-ss-ERp44

---

structure of TMX5. Cysteine residues are in green, and the active site cysteine 220 is in red. The proximity of cysteine residues 177 and 210 in the TMX5 structure suggests their engagement in an intramolecular disulfide bond (https://alphafold.ebi.ac.uk/entry/Q96J42). **(D)** Co-localization in CLSM of TMX5 with the ER marker CNX. **(E)** Same as (D) for $TMX5_{C220A}$. **(F)** Same as (D) for $TMX5_{C114A}$. **(G)** Same as (D) for $TMX5_{C124A}$. **(H)** Same as (D) for $TMX5_{C177A}$. **(I)** Same as (D) for $TMX5_{C210A}$. **(J)** Co-localization in CLSM of TMX5 with the trans-Golgi marker Giantin. **(K)** Same as (J) for $TMX5_{C220A}$. **(L)** Same as (J) for $TMX5_{C114A}$. **(M)** Same as (J) for $TMX5_{C124A}$. **(N)** Same as (J) for $TMX5_{C177A}$. **(O)** Same as (J) for $TMX5_{C210A}$. Scale bars: 10 $\mu$m.
Source data are available for this figure.

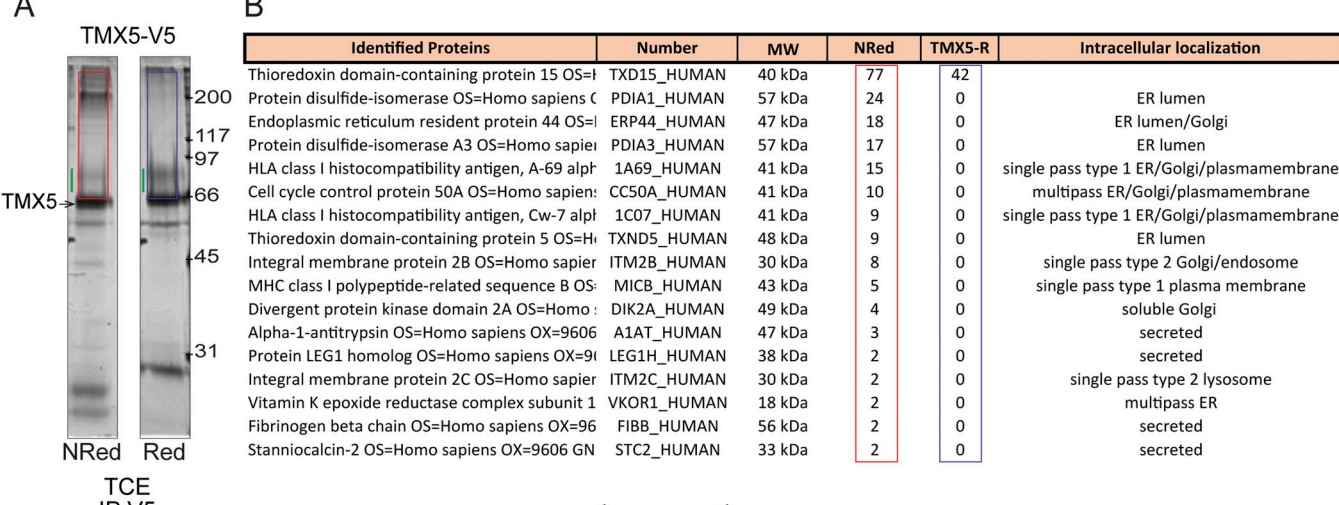

**A**

TMX5-V5

TMX5 →

NRed  Red

TCE
IP:V5
Silver stained

200
117
97
66
45
31

**B**

| Identified Proteins | Number | MW | NRed | TMX5-R | Intracellular localization |
|---|---|---|---|---|---|
| Thioredoxin domain-containing protein 15 OS=H | TXD15_HUMAN | 40 kDa | 77 | 42 | |
| Protein disulfide-isomerase OS=Homo sapiens C | PDIA1_HUMAN | 57 kDa | 24 | 0 | ER lumen |
| Endoplasmic reticulum resident protein 44 OS=I | ERP44_HUMAN | 47 kDa | 18 | 0 | ER lumen/Golgi |
| Protein disulfide-isomerase A3 OS=Homo sapien | PDIA3_HUMAN | 57 kDa | 17 | 0 | ER lumen |
| HLA class I histocompatibility antigen, A-69 alph | 1A69_HUMAN | 41 kDa | 15 | 0 | single pass type 1 ER/Golgi/plasmamembrane |
| Cell cycle control protein 50A OS=Homo sapiens | CC50A_HUMAN | 41 kDa | 10 | 0 | multipass ER/Golgi/plasmamembrane |
| HLA class I histocompatibility antigen, Cw-7 alph | 1C07_HUMAN | 41 kDa | 9 | 0 | single pass type 1 ER/Golgi/plasmamembrane |
| Thioredoxin domain-containing protein 5 OS=H | TXND5_HUMAN | 48 kDa | 9 | 0 | ER lumen |
| Integral membrane protein 2B OS=Homo sapier | ITM2B_HUMAN | 30 kDa | 8 | 0 | single pass type 2 Golgi/endosome |
| MHC class I polypeptide-related sequence B OS= | MICB_HUMAN | 43 kDa | 5 | 0 | single pass type 1 plasma membrane |
| Divergent protein kinase domain 2A OS=Homo : | DIK2A_HUMAN | 49 kDa | 4 | 0 | soluble Golgi |
| Alpha-1-antitrypsin OS=Homo sapiens OX=9606 | A1AT_HUMAN | 47 kDa | 3 | 0 | secreted |
| Protein LEG1 homolog OS=Homo sapiens OX=9( | LEG1H_HUMAN | 38 kDa | 2 | 0 | secreted |
| Integral membrane protein 2C OS=Homo sapier | ITM2C_HUMAN | 30 kDa | 2 | 0 | single pass type 2 lysosome |
| Vitamin K epoxide reductase complex subunit 1 | VKOR1_HUMAN | 18 kDa | 2 | 0 | multipass ER |
| Fibrinogen beta chain OS=Homo sapiens OX=96 | FIBB_HUMAN | 56 kDa | 2 | 0 | secreted |
| Stanniocalcin-2 OS=Homo sapiens OX=9606 GN | STC2_HUMAN | 33 kDa | 2 | 0 | secreted |

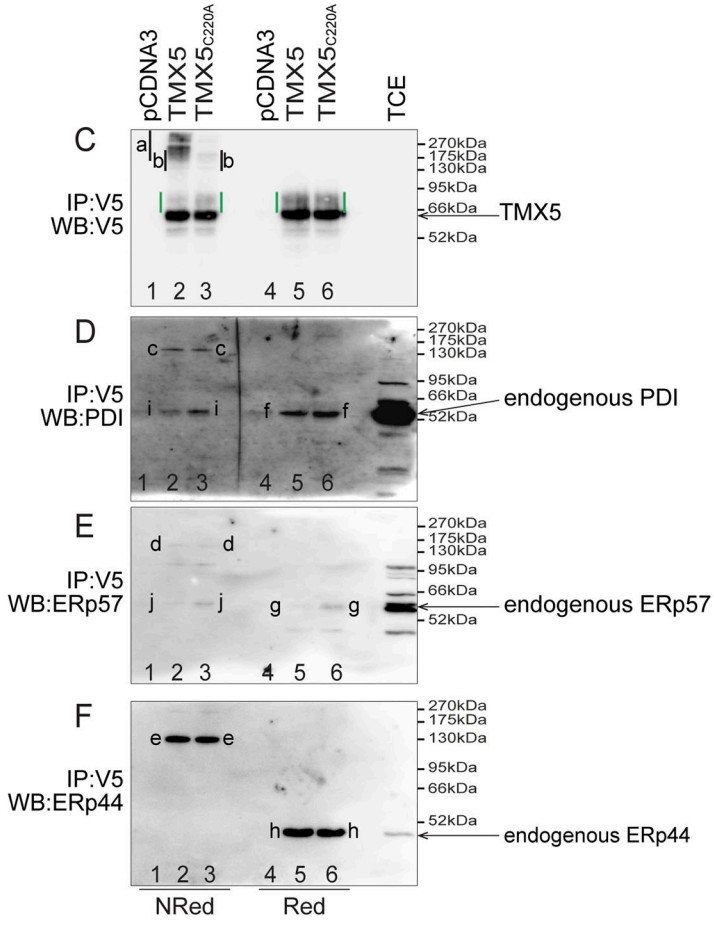

**C**

IP:V5
WB:V5

270kDa
175kDa
130kDa
95kDa
66kDa → TMX5
52kDa

1 2 3    4 5 6

**D**

IP:V5
WB:PDI

270kDa
175kDa
130kDa
95kDa
66kDa → endogenous PDI
52kDa

1 2 3    4 5 6

**E**

IP:V5
WB:ERp57

270kDa
175kDa
130kDa
95kDa
66kDa
52kDa → endogenous ERp57

1 2 3    4 5 6

**F**

IP:V5
WB:ERp44

270kDa
175kDa
130kDa
95kDa
66kDa
52kDa → endogenous ERp44

1 2 3    4 5 6

NRed        Red

**Figure 3.  Endogenous interactors of TMX5.**
**(A)** V5-tagged TMX5 has been immunoisolated from HEK293 cell lysates. Part of the immunocomplexes were separated in non-reducing/reducing gel and stained with silver. The endogenous proteins engaged in mixed disulfides with TMX5 (red rectangle) have been identified by LC/MS analyses after trypsin digestion in gel (two independent experiments). **(B)** Select endogenous polypeptides engaged by TMX5 in mixed disulfides are shown in the red rectangle. **(C)** Engagement of TMX5 clients in mixed disulfides (lane 2) is substantially inhibited upon replacement of the active site $Cys_{220}$ with alanine (lane 3). *a* shows TMX clients engaged by the active site $Cys_{220}$; *b* shows interaction partners, whose covalent association persists upon mutation of the active site cysteine; and green lines show complex glycosylated TMX5. **(D)** *c* shows the endogenous PDI component of TMX5 complexes immunoisolated with anti-V5 antibodies. Upon reduction of the sample, the disulfide-bonded complexes are disassembled and the PDI component is shown with *f*. PDI is engaged in mixed disulfides both with TMX5 (lane 2) and with $TMX5_{C220A}$ (lane 3). Note that PDI also associates non-covalently with TMX5 and $TMX5_{C220A}$ (shown with *i*). **(E)** Same as (D) for ERp57. The ERp57 component of TMX5 mixed disulfides is shown with *d*, ERp57

complexes in cells expressing TMX5 and ERp44 (Fig 5A, lanes 2 and 5) but not in cells expressing TMX5 and ERp44$_{C29S}$ (lanes 3 and 6). Analyses of the electrophoretic mobility of TMX5 (Fig 5A, lanes 7–12, green lines) and the EndoH assay (Fig 5B and quantification in Fig 5C) also reveal that the association with ERp44 retains TMX5 in the ER. This is shown by the substantial reduction in TMX5 complex glycosylation from about 20% EndoH-resistant fraction in mock-transfected cells (Figs 2B and 5B, lanes 1, 2) and in cells expressing ERp44$_{C29S}$ (Fig 5B, lanes 5 and 6), to less than 10% upon the co-expression of ERp44 (Fig 5B, lanes 3 and 4). Replacement of the active site TMX5 cysteine 220 with alanine abolishes the engagement of clients in mixed disulfides (Fig 3C, lanes 3 and 4; and Fig 5A, lanes 4–6). This mutation does not affect the fraction of TMX5 that acquires EndoH-resistant N-glycans as a symptom of ER-to-Golgi trafficking (Fig 5B, lanes 7 and 8; and Fig 5C), does not prevent engagement of ERp44 (Fig 3F), nor the capacity of ERp44 to inhibit release of TMX5 from a pre-medial Golgi compartment (Fig 5B, lanes 9–10; and Fig 5C). The role of ERp44 in determining the intracellular localization of TMX5 has been confirmed by CLSM showing that TMX5 and TMX5$_{C220A}$ are excluded from the Giantin-positive compartment when co-expressed with ERp44 (Fig 5E and H), but traffic to the Giantin-positive compartment when expressed alone (Fig 5D and G) or when co-expressed with the active site mutant form ERp44$_{C29S}$ (Fig 5F and I).

Notably, the co-expression of ERp44 substantially reduces the fraction of TMX5 engaged with endogenous clients in mixed disulfides (compare Fig 5A, lanes 1 versus 2, shown with *a*). The co-expression of the ERp44$_{C29S}$ mutant that does not engage TMX5 in covalent complexes does not affect the formation of mixed disulfides between TMX5 and its clients (Fig 5A, lane 3). One possible interpretation of these data is that efficient ER retention of TMX5 partially impairs its association with clients. Another is that the covalent association of ERp44 competes with client association (even if ERp44 and clients engage different TMX5 cysteine residues) or facilitates the release of clients by TMX5.

### Knockout of ERp44 substantially impairs retention of TMX5 in pre-Golgi compartments

To further confirm the role of ERp44 in the retention of TMX5 in pre-Golgi compartments, TMX5 was transiently transfected in ERp44 knockout HeLa cells (Fig 6A). TMX5 was expressed alone (Fig 6B, lanes 1 and 4) or was co-expressed with ERp44 (lanes 2 and 5) or with ERp44$_{C29S}$, the form of ERp44 that does not associate with TMX5 (lanes 3 and 6). The EndoH assay performed on the TMX5 expressed in cells lacking ERp44 revealed that about 50% of the protein acquires EndoH-resistant N-glycans (Fig 6C, lane 2). In wild-type cells, this value was around 20% (Figs 2B and 5B and C). Notably, the co-transfection of ERp44 substantially reduces the fraction of TMX5 displaying complex glycans to 29% (lane 4), whereas the co-transfection of ERp44$_{C29S}$ does not compensate the lack of ERp44 in retaining the polypeptide in pre-Golgi compartments.

### The covalent association of ERp44 with TMX5 relies on TMX5's cysteine residues 114 and 124

Finally, we establish the cysteine residue(s) of TMX5 that form(s) the covalent disulfide bond with the active site Cys$_{29}$ of ERp44, aware that the catalytic Cys$_{220}$ of TMX5 is not the one (Fig 3F). To this end, HA-tagged ERp44 was co-expressed with individual V5-tagged cysteine mutants of TMX5 in HEK293 cells (Fig 7A). The formation of covalent complexes between ERp44 and TMX5 variants was tested upon immunoisolation of the complexes with anti-HA antibodies, separation of the immunocomplexes in non-reducing/reducing gel, and detection of the TMX5-interacting partner with anti-V5 antibodies. As shown in Fig 3F, ERp44 engages in mixed disulfides with TMX5 (Fig 7B, lane 1) and TMX5$_{C220A}$ (Fig 7B, lane 2). The individual mutation of Cys$_{114}$ or of Cys$_{124}$ to alanine residues reduces, without abolishing, the covalent engagement, ERp44 (Fig 7B, lanes 3 and 4). Mutations of Cys$_{177}$ (lane 5) and of Cys$_{210}$ (lane 6) allow the covalent engagement of TMX5 with formation of aberrant complexes possibly resulting from TMX5 misfolding/instability because of the deletion of the structural disulfide that binds these two cysteine residues (red lines, lanes 5 and 6). The results were confirmed upon immunoisolation of the TMX5 component of the mixed disulfide and a Western blot that reveals the presence of ERp44 in the immunocomplexes (Fig 7C). Thus, the active site Cys$_{29}$ of ERp44 engages Cys$_{114}$ and Cys$_{124}$ of TMX5 in mixed disulfides to retain TMX5 in a pre-medial Golgi complex.

## Discussion

Our study offers a preliminary set of information on a neglected member of the PDI family by showing that TMX5 engages a subset of cellular proteins in mixed disulfides. Most of these covalent interactions occur via the active site cysteine 220 of TMX5 and likely control transient or long-lasting association of TMX5 with clients to determine TMX5's biological activity that is the focus of a follow-up study ongoing in the laboratory. Notably, the set of endogenous proteins that engage TMX5 in mixed disulfides is distinct from the endogenous proteins engaged in mixed disulfides by TMX1 (10), TMX3 (17), and TMX4 (17). This confirms the exquisite selectivity of individual members of the PDI superfamily, which also applies to the membrane-tethered TMX proteins, and results from their distinct substrate interaction domains, different co-factors, and different subcompartmental distribution (17, 32). Silencing or knockout of TMX5 turns out to be lethal for cultured cells in our attempts, which seems consistent with the perinatal lethality of inherited TMX5 mutations (22, 23, 24).

The members of the protein disulfide isomerase superfamily PDI (PDIA1), ERp57 (PDIA3), and ERp44 are major interactors of TMX5. They form a mixed disulfide with one or more cysteine residues outside the TMX5 active site. Thus, they are not *clients* of TMX5; if one defines as clients of an oxidoreductase, those endogenous

---

associated non-covalently with TMX5 is shown with *j*, and endogenous ERp57 disassembled from the covalent complexes with TMX5 upon sample reduction is shown with *g*. **(F)** Same as (D) for ERp44. The ERp44 component of TMX5 mixed disulfides is shown with *e*, and endogenous ERp44 disassembled from the covalent complexes with TMX5 upon sample reduction is shown with *h*.

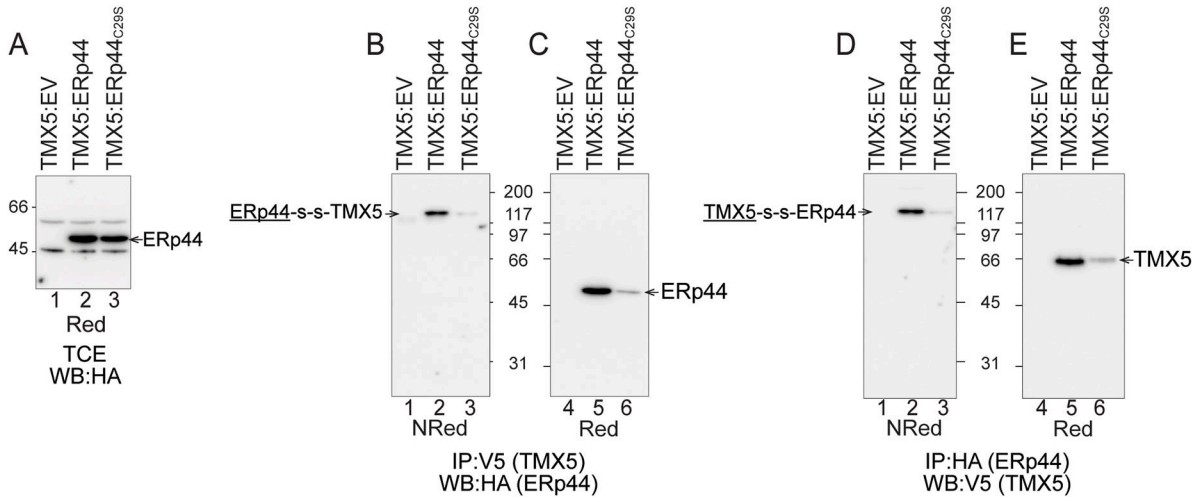

**Figure 4. TMX5 engages ERp44 in mixed disulfides via its catalytic Cys29.**
**(A)** WB of the total cell extract to show the level of ERp44 in cells transfected for the expression of TMX5-V5 (lane 1), TMX5-V5 and ERp44-HA (lane 2), or TMX5-V5 and ERp44$_{C29S}$-HA (lane 3). Samples are analyzed under reducing conditions (Red). **(B)** Formation of mixed disulfides between TMX5 and ERp44 (lane 2) or ERp44$_{C29S}$ (lane 3). Mixed disulfides are immunoisolated with anti-V5 antibodies and are separated in NRed SDS–PAGE. The ERp44 component (underlined) of the ERp44-s-s-TMX5 mixed disulfide is revealed by WB with an anti-HA antibody. **(B, C)** Same as (B), where the mixed disulfides are disassembled under reducing conditions. The ERp44 polypeptide component of the mixed disulfide is shown. **(B, D)** Same as (B), where the mixed disulfides have been immunoisolated from the detergent extract with an anti-HA antibody and the TMX5 component (underlined) is revealed in WB with an anti-V5 antibody. **(C, E)** Same as (C) for TMX5.

polypeptides transiently or permanently associate with the oxidoreductase's active site cysteine residues. Rather, they could play a role in ruling the biological activity of TMX5 by acting as co-factors. PDI and ERp57 oxidoreductases could regulate the release of clients from TMX5. In fact, the N-terminal cysteine of the TMX5 active site can nucleophilically attack a free thiol group in protein substrates, but the interactions can be resolved only by the intervention of an external cysteine provided by glutathione, or by another PDI (21). This hypothesis will be verified in the ongoing follow-up study aiming at establishing the physiologic role of TMX5 by monitoring the fate of model clients as we did previously for TMX1 (10, 11) and TMX4 (4, 17, 18).

The results presented in this work allow to add TMX5 to the limited group of proteins including Ero1α, Ero1β, Prx4, ERAP1, and SUMF1 that rely on ERp44 engagement to control their subcellular localization. Notably, in contrast to the other ERp44 clients, TMX5 is tethered to the ER membrane. Why these proteins that exert most of their known activities in the ER or in pre-Golgi compartments do not rely on conventional C-terminal retention/ retrieval sequences to be retained in the compartment is a matter of discussion. It is likely that under specific physiologic or pathologic circumstances, the engagement of ERp44 can be modulated to allow ERp44 clients to exert their action in the Golgi compartment, at the cell surface, or extracellularly (30). It should be noted that in the case of TMX5, the extensive N-glycosylation could engage leguminous L-type lectins located in the ER (VIPL), cycling between the ER and the intermediate compartment (ERGIC) (ERGIC-53) or between the ERGIC and the *cis*-Golgi (VIP36) (33, 34, 35, 36), and have an impact on the subcellular distribution and activity of TMX5.

In addition to ERp44 and TMX5, other members of the protein disulfide isomerase superfamily contain an incomplete thioredoxin domain, including ERp29, PDIA17 (AGR3), and PDIA18 (AGR2), some of which are expressed in specific tissues. The absence of one cysteine residue in the active site prolongs the association with clients by stabilizing mixed disulfides that are normally extremely short-living and challenging to be investigated in cellular localization (37). It facilitates the identification of the endogenous proteins that engage the oxidoreductase's catalytic site in long-living mixed disulfides, as shown in this work for TMX5. With the relevant exception of ERp44 (21, 25, 29, 30, 31), the physiologic function of these "holdases" is poorly understood. Recently, the goblet-specific PDIA18/AGR2 protein has been involved in preventing activation of the unfolded protein response by associating with the unfolded protein response sensor IRE1β thereby impeding dimerization, even if it remains questionable if this function involves the formation of disulfide bonds via the AGR2 incomplete thioredoxin site with its client(s). A direct covalent association between AGR2 and IRE1β has been excluded (38, 39, 40).

Biologists have learned to use the peculiar feature of holdases, and the mutation of the last cysteine residue of the oxidoreductase's CXXC catalytic sites is an established protocol to generate *artificial trapping mutants* that are instrumental for the identification of the oxidoreductase's clients upon co-immunoprecipitation and mass spectrometry analyses (41, 42, 43). The identity of the clients may lead to infer the oxidoreductase's physiologic function. Referring to other TMX family members studied in our laboratory (4), the trapping mutant approach led us to discover that TMX1 preferentially associates with membrane-bound polypeptides and selectively controls ER-associated degradation of membrane-associated folding-defective proteins by acting as a reductase (10, 11) and that TMX4 preferentially associates with nuclear envelope polypeptides and controls redox

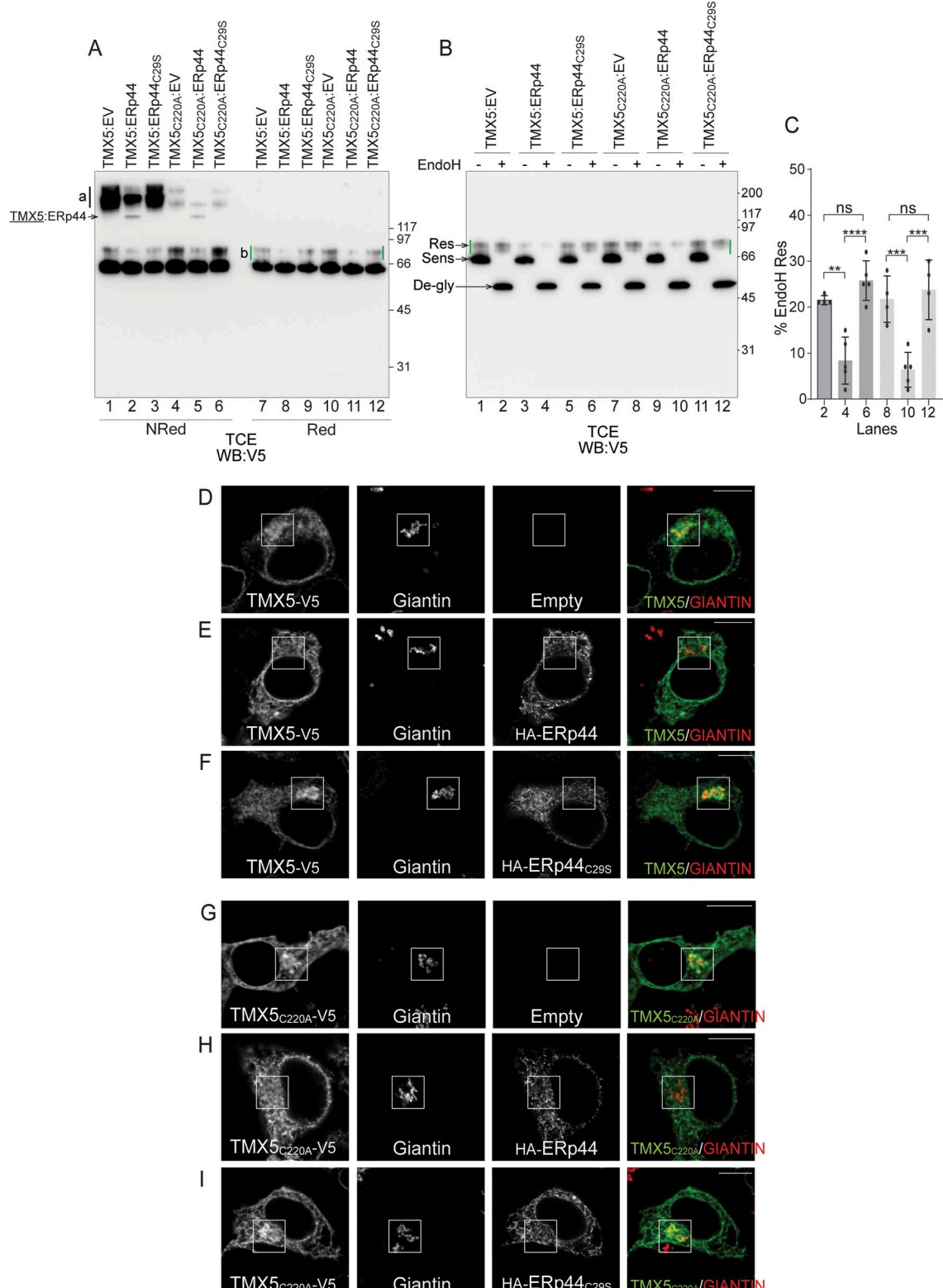

**Figure 5. Association with ERp44 retains TMX5 in the ER and reduces engagement of endogenous clients.**
**(A)** Western blot analysis of post-nuclear fractions of HEK293 cells co-transfected with V5-tagged TMX5 and HA-tagged ERp44 or ERp44$_{C29S}$. The TMX component of the complexes is revealed with anti-V5 antibody in non-reducing and reducing conditions (lanes 1–6 and 7–12, respectively). Mixed disulfides between TMX5 and endogenous clients are shown with the black line *a*. TMX5:ERp44 mixed disulfides are shown in lanes 2 and 5. The complex glycosylated TMX5 is shown with the green lines *b*. **(B)** EndoH assay of the same samples. The EndoH-resistant, Golgi forms of TMX5 are shown with green lines. The EndoH-resistant, EndoH-sensitive, and EndoH–de-glycosylated

reactions that disassemble the LINC complexes that establish the distance between inner and outer nuclear membranes (4, 17, 18). In the case of TMX5, we note that HLA class I proteins are the major covalent interactors of TMX5, besides ERp57, PDI, and ERp44 (Fig 3B). It may not be a coincidence that mutations in the *TMX5* gene are associated with the development of the rare, perinatally lethal, Meckel–Gruber syndrome caused by defective ciliogenesis (22, 23, 24) and that the transcription factors of the RFX family that control HLA class I expression (44, 45, 46) have been implicated in cilium-associated functions in zebrafish, Xenopus, and mammals (47, 48, 49).

# Materials and Methods

## Cell culture and transient transfection

HEK293, MEFs, and HeLa wild-type and ERp44-KO (a kind gift from Roberto Sitia) cell lines were cultured at 37°C and 5% $CO_2$ in DMEM with high glucose (GlutaMAX, Gibco) supplemented with 10% FBS (Gibco). Transient transfections were carried out using jetPRIME (Polypus) in DMEM/10% FBS supplemented with non-essential amino acids (Gibco) following the manufacturer's protocol. Experiments were performed 17 h after transfection.

## Antibodies and expression plasmids

Table S1 shows the details of the antibodies used. Plasmids encoding for TMX5-V5 WT and cysteine mutants were synthesized by GenScript in pUC57 and subcloned in pcDNA3.1(–) through restriction digestion of BamHI and HindIII (NEB) sites flanking the target sequence. Ligation was carried out using T4 DNA ligase (NEB) at a 3:1 insert-to-vector ratio, following the manufacturer's protocol. Plasmids were subsequently amplified and isolated from JM109 bacteria (Promega) using the GenElute HP Plasmid MidiPrep kit (Sigma-Aldrich). Plasmids for the expression of ERp44wt and C29S mutants were a kind gift from Roberto Sitia.

## Cell lysis, immunoprecipitation (IP), Western blot (WB), and EndoH treatment

HEK293 cells were washed with ice-cold 1xPBS containing 20 mM N-ethylmaleimide (NEM), then lysed with RIPA buffer containing 20 mM NEM and protease inhibitors (1 mM PMSF, 16.5 mM chymostatin, 23.4 mM leupeptin, 16.6 mM antipain, and 14.6 mM pepstatin) for 20 min on ice. Lysates were subjected to centrifugation for 10 min at 4°C, 10,600*g* to extract the post-nuclear supernatant. For immunoprecipitation, post-nuclear supernatants were incubated with Anti-V5 Agarose Affinity Gel–conjugated beads or with protein A beads (1:10 wt/vol swollen in

PBS; Sigma-Aldrich) and anti-HA antibody for 2 h at 4°C. Samples were then washed with 1% Triton in HBS 1x, pH 7.4, and then dried. Dried beads were resuspended in sample buffer and boiled at 95°C for 5 min. After boiling, samples without (non-reducing) or with (reducing) 100 mM DTT (Roche) were subjected to SDS–PAGE. For detection of proteins by Western blot (WB), protein bands (IP or TCE) were transferred from polyacrylamide gel to a PVDF membrane using a Trans-Blot Turbo device (Bio-Rad). A PVDF membrane was blocked for 10 min with 8% blocking milk in Tris-buffered saline, 0.1% Tween-20 (vol/vol) (TBS-T) and incubated with primary antibodies (Table S1) with shaking. After primary antibody washout with TBS-T, membranes were incubated with HRP-conjugated secondary antibody or with HRP-conjugated protein A (Table S1) for 45 min at RT while shaking. Protein bands were detected on a Fusion FX7 chemiluminescence detection system (Vilber) using WesternBright Quantum or WesternBright ECL HRP Substrate (Witec AG) following the manufacturer's protocol. Protein bands were quantified with FIJI/ImageJ software. For EndoH (NEB) treatment, proteins from immunoprecipitated samples or from TCE were split into two aliquots and incubated in the presence or absence of 5 mU of EndoH for 3 h at 37°C according to the manufacturer's protocol. Samples were then analyzed by SDS–PAGE.

## Mass spectrometry

HEK293 cells mock-transfected (pcDNA3.1) or expressing TMX5-V5 were lysed in RIPA buffer. The post-nuclear supernatant was double-immunoprecipitated using anti-V5–conjugated beads. After washing, the beads were extracted by boiling for 5 min in 60 $\mu l$ of sample buffer. The supernatant was split into two aliquots, one of which was reduced by adding DTT to a final concentration of 10 mM. Reduced and non-reduced samples were separated in a gel that was silver-stained (Fig 3A). The polypeptides present in the red (non-reducing lane) and blue (reducing lane) rectangles were digested in-gel as described previously (50, 51). Extracted tryptic peptides were dried and resuspended in 0.05% trifluoro-acetic acid, 2% (vol/vol) acetonitrile. MS analysis was performed at the Protein Analysis Facility, University of Lausanne, Switzerland.

## CLSM

HEK cells were seeded on poly-L–coated (Sigma-Aldrich) glass coverslips (VWR) and transiently transfected as indicated using jetPRIME (Polypus) in DMEM/10% FBS supplemented with non-essential amino acids (Gibco). Seventeen hours after transfection, cells were fixed at RT for 20 min in 3.7% formaldehyde (FA, vol/vol) in PBS. Coverslips were incubated for 20 min in permeabilization solution (PS, 10 mM Hepes, 15 mM glycine, 10% goat serum [vol/vol], and 0.05% saponin [wt/vol]). After permeabilization, primary

---

TMX5 are shown with arrows. **(B, C)** Quantification of the EndoH-resistant TMX5 in the indicated lanes of the gel in panel (B). n = 4, 5, 5, 4, 5, 4 biological replicates, respectively. Ordinary one-way ANOVA, F = 15.78, ****$P < 0.0001$, ***$P < 0.001$, **$P < 0.01$, ns: not significant ($P > 0.05$). Data are presented as the mean + SD. **(D)** Partial co-localization of TMX5-V5 with the Golgi marker Giantin in mock-transfected cells. **(E)** Lack of co-localization of TMX5 with Giantin upon the co-expression of ERp44. **(F)** Partial co-localization of TMX5 with Giantin upon the co-expression of ERp44$_{C29S}$. **(D, E, F, G, H, I)** Same as (D, E, F) for TMX5$_{C220A}$. Scale bars: 10 $\mu m$.

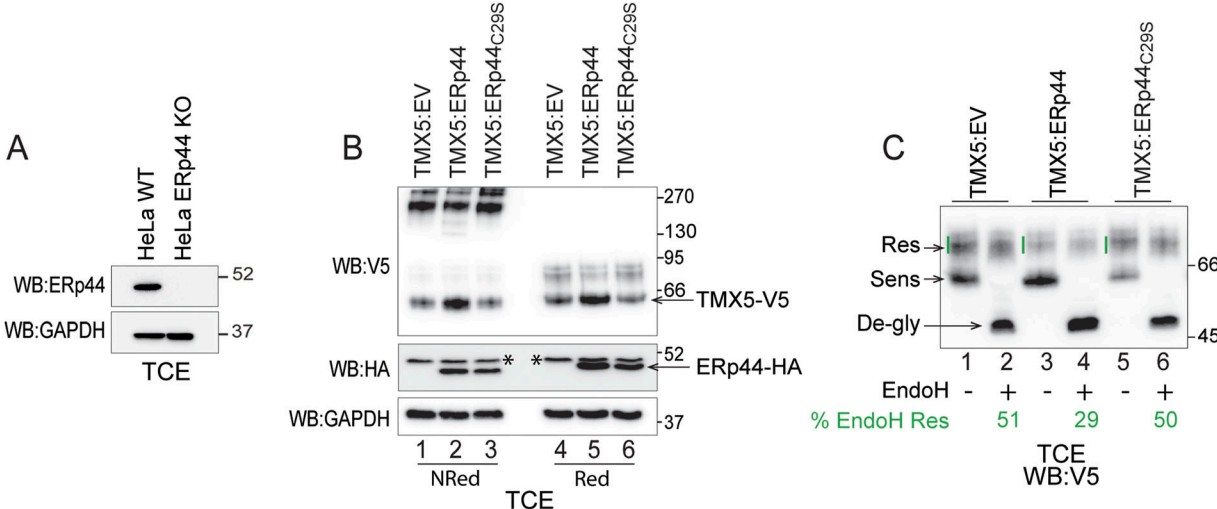

**Figure 6. ERp44 knockout substantially impairs pre-Golgi retention of TMX5.**
**(A)** Western blot showing the knockout of ERp44. **(B)** Expression of TMX5 (upper panel, non-reducing gel [lanes 1–3] and reducing gel [lanes 4–6]), of recombinant ERp44 (middle panel, lanes 2 and 5), and of ERp44$_{C29S}$ (middle panel, lanes 3 and 6). Asterisks show a cross-reacting band used as a loading control. GAPDH is shown as an additional loading control. **(C)** Assessment of EndoH-resistant N-glycans of TMX5 expressed alone (lanes 1, 2), or co-expressed with ERp44 (lanes 2 and 4) or with ERp44$_{C29S}$ (lanes 3 and 6). The EndoH-resistant, EndoH-sensitive, and EndoH–de-glycosylated TMX5 are shown with arrows. Values for this experiment are shown in green.

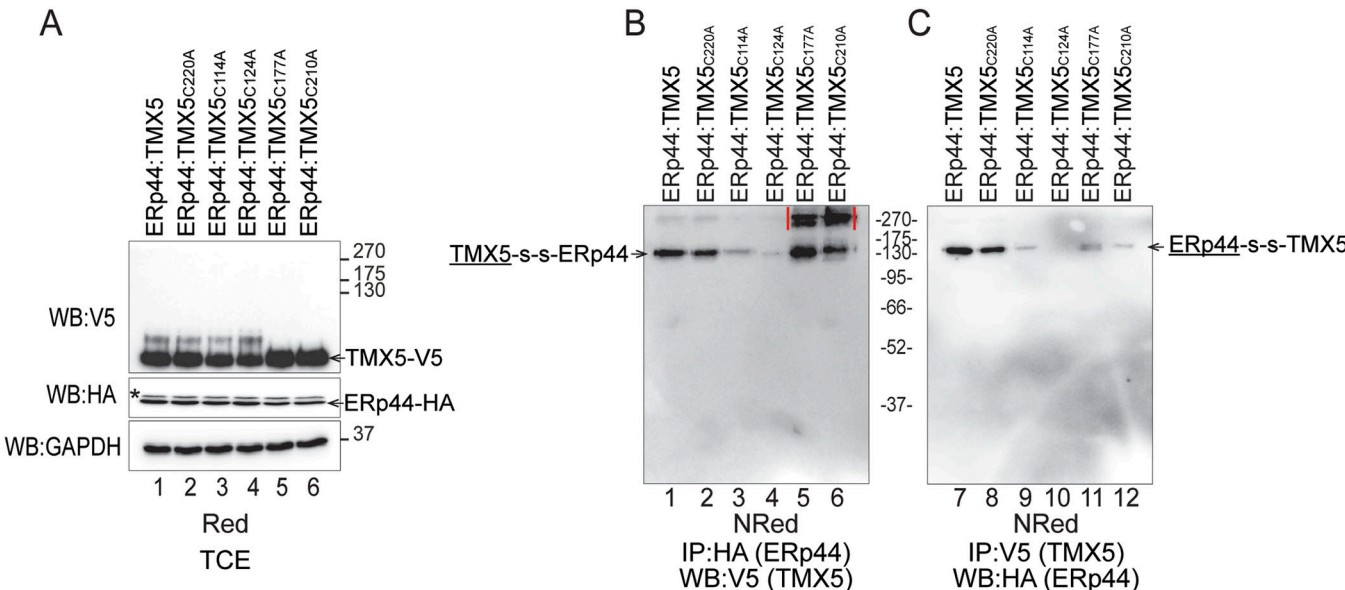

**Figure 7. ERp44 engages Cys$_{114}$ and/or Cys$_{124}$ of TMX5 in mixed disulfides.**
**(A)** Western blots showing the expression of TMX5 variants (upper panel) and of ERp44 (middle panel). GAPDH is shown as an additional loading control. **(B)** Co-precipitation of cysteine mutants of TMX5 with ERp44. The mixed disulfides are separated in a non-reducing gel. Here, the TMX5 component of the ERp44-S-S-TMX5 complex is shown. **(C)** Co-precipitation of ERp44 with TMX5. The ERp44 component of the ERp44-S-S-TMX5 complex is shown in the non-reducing gel.

antibodies (Table S1) diluted 1:100 in PS were applied for 2 h, washed three times in PS, and then incubated with Alexa Fluor–conjugated secondary antibodies diluted 1:300 in PS for 45 min. Cells were washed three times with PS and once with deionized water and mounted on a drop of VECTASHIELD (Vector Laboratories) supplemented with 40, DAPI. Coverslips were imaged using a Leica TCS SP5 microscope equipped with a Leica HCX

PL APO lambda blue 63.0 × 1.40 oil objective. Leica LAS X software was used for image acquisition, with excitation provided by 488-, 561-, and 633-nm lasers, and fluorescence light was collected within the ranges of 504–587 nm (Alexa Fluor 488), 557–663 nm (TMR), and 658–750 nm (Alexa Fluor 646), respectively, with pinhole 1 AU. Image post-processing was performed with Adobe Photoshop.

## Data Availability

All data, code, and materials used in the analysis are available, without restriction upon reasonable request to reproduce or extend the analyses. The mass spectrometry proteomics data have been deposited to the ProteomeXchange Consortium via the PRIDE (52) partner repository with the dataset identifiers PXD054716.

## Supplementary Information

## Acknowledgements

We thank the members of Molinari's laboratory for discussions and critical reading of the article. This study was supported by Swiss National Science Foundation grants 310030_214903 and 320030-227541 (to M Molinari).

### Author Contributions

T Soldà: conceptualization, data curation, formal analysis, validation, investigation, visualization, methodology, and writing—review and editing.
C Galli: conceptualization, data curation, formal analysis, validation, investigation, visualization, methodology, and writing—review and editing.
C Guerra: conceptualization and investigation.
C Hoefner: investigation.
M Molinari: conceptualization, data curation, supervision, validation, project administration, and writing—original draft.

### Conflict of Interest Statement

The authors declare that they have no conflict of interest.

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
