## [Reviewer comments · Life Science Alliance]

Life Science Alliance

TMX5/TXNDC15, a natural trapping mutant of the PDI family is client of the proteostatic factor ERp44

Tatiana Soldà, Carmela Galli, Concetta Guerra, Carolin Hoefner, and Maurizio Molinari

DOI: 10.26508/lsa.202403047

Corresponding author(s): Maurizio Molinari, Institute for Research in Biomedicine

Review Timeline:

Submission Date:	2024-09-17
Editorial Decision:	2024-09-19
Revision Received:	2024-09-23
Accepted:	2024-09-23

Transaction Report:

Please note that the manuscript was reviewed at Review Commons and these reports were taken into account in the decision-making process at Life Science Alliance.

Review
COMMONS

September 19, 2024

RE: Life Science Alliance Manuscript #LSA-2024-03047

Prof. Maurizio Molinari
Institute for Research in Biomedicine
Protein Folding and Quality Control
Via V. Vela 6
Bellinzona CH-6500
Switzerland

Dear Dr. Molinari,

Thank you for submitting your revised manuscript entitled "TMX5/TXNDC15, a natural trapping mutant of the PDI family is a client of the proteostatic factor ERp44". We would be happy to publish your paper in Life Science Alliance pending final revisions necessary to meet our formatting guidelines.

- please be sure that the authorship listing and order is correct
- please upload your figures as single files
- please add a summary blurb/alternate abstract and a category for your manuscript to our system
- please add the Twitter handle of your host institute/organization as well as your own or/and one of the authors in our system
- please use the [10 author names, et al.] format in your references (i.e. limit the author names to the first 10)
- please add a separate figure legend section to the manuscript

Figure Check:

- For Figure 2, please make sure that all panels are referred to in the figure legend
- please provide the original blots as source data for Figure 2B
- please add scale bars to Figure 5 D-I

LSA now encourages authors to provide a 30-60 second video where the study is briefly explained. We will use these videos on social media to promote the published paper and the presenting author (for examples, see <https://docs.google.com/document/d/1-UWCfbE4pGcDdcgzcmiuJl2XMBJnxKYeqRvLLrLS08s/edit?usp=sharing>). Corresponding or first-authors are welcome to submit the video. Please submit only one video per manuscript. The video can be emailed to contact@life-science-alliance.org

A. FINAL FILES:

B. MANUSCRIPT ORGANIZATION AND FORMATTING:

Sincerely,

September 23, 2024

RE: Life Science Alliance Manuscript #LSA-2024-03047R

Prof. Maurizio Molinari
Institute for Research in Biomedicine
Protein Folding and Quality Control
Via V. Vela 6
Bellinzona CH-6500
Switzerland

Dear Dr. Molinari,

Thank you for submitting your Research Article entitled "TMX5/TXNDC15, a natural trapping mutant of the PDI family is client of the proteostatic factor ERp44". It is a pleasure to let you know that your manuscript is now accepted for publication in Life Science Alliance. Congratulations on this interesting work.

DISTRIBUTION OF MATERIALS:

Again, congratulations on a very nice paper. I hope you found the review process to be constructive and are pleased with how the manuscript was handled editorially. We look forward to future exciting submissions from your lab.

Sincerely,
